# Atmospheric wavenumber-4 driven South Pacific marine heat waves and marine cool spells

Stephen M. Chiswell ⬡ [1]✉

Marine heat waves (MHW) and cool spells (MCS) can both positively and negatively impact marine ecosystems with potentially large societal and economic impacts. Here, I examine the global teleconnections of MHW/MCS in the southern hemisphere and Tasman Sea. When MHW/MCS are defined with respect to a linear warming trend, there is little evidence that MHW in the Tasman Sea are changing in either frequency or intensity but may be lasting longer. MCS may be becoming weaker and less frequent. I show that MHW/MCS in the Tasman Sea co-occur with corresponding events in the Atlantic, Indian, and eastern-Pacific Oceans, and these southern hemisphere events are likely driven by stalling of a global wavenumber-4 (W4) atmospheric wave, leading to anomalously weak north-easterly winds during MHW or strong south-westerly winds during MCS. Thus, the key to predicting MHW/MCS is in understanding what causes the atmospheric W4 wave to stall.

[1] National Institute of Water and Atmospheric Research, Wellington, New Zealand. ✉email: s.chiswell@niwa.cri.nz

It has been suggested that marine heat waves (MHW) have become stronger and or more frequent over the last century[1,2] (although it has been pointed out that such suggestions may not take into account long-term warming[3]). MHW are sometimes considered as good analogues for possible future oceans[4], and are often associated with detrimental impact on ocean primary production, for example, extreme MHW have been associated with a loss of kelp forests in Western Australia[5], a massive mortality of sea birds in the northeast Pacific Ocean[6], and a collapse of the salmon fishing industry around New Zealand[7].

A recent review of MHW[8] shows that despite considerable interest and research into them, there are still many unanswered questions, for example, whether they are locally or distantly forced, what the relative roles of atmospheric and oceanic forcing are, and the degree of similarity between different MHW. Some research has suggested that not all MHW are driven by the same mechanisms, for example, Holbrook et al.[9] showed a variety of forcing mechanisms led to a heterogeneous distribution of occurrence and duration around the globe. In contrast, Sen Gupta et al.[10] suggested that globally, almost all extreme MHW were associated with reduced wind speeds during their build up phase. In some, but not all, cases this was also related to suppressed turbulent heat losses from the ocean (in particular, latent heat). Bond et al.[11] suggested MHW in the northeast Pacific Ocean were primarily driven by reduced ocean mixing associated with a persistent ridge of higher-than-normal atmospheric pressure.

In the southern hemisphere, there have been a variety of proposed mechanisms for MHW. Behrens et al.[12] suggested that heat content fluctuations in the Tasman Sea are predominately controlled by variations in the meridional heat transport from the subtropics via the East Australian Current, impacted by wind stress curl anomalies north of the region. Li et al.[13] similarly suggested about half of historical Tasman Sea MHW were due to increased poleward transport within the East Australian Current, but that the variability is driven by westward-propagating sea surface height anomalies from the interior South Pacific. There are also questions about how deep MHW penetrate, whether they are simply due to a lack of wind stress leading to surface-intensified warming[4], or whether they can show deep expression with maximum warming below the surface as seen off Western Australia[14].

Recently, there has been a suggestion that southern-hemisphere SST variability is globally interconnected. Senapati et al.[15] revealed the presence of a stationary zonal wavenumber-4 (W4) pattern in SST anomaly in the southern subtropics (20°S–55°S), i.e., where SST anomalies showed four highs spaced about 90° of longitude apart, south of Australia, central Pacific, western Atlantic and western Indian Oceans, interspersed with four lows—see their Fig. 1. This pattern is seasonally phase-locked to the austral summer and persists up to mid-autumn. They suggested that thermodynamic coupling of the atmosphere and the upper ocean helps in generating the W4 pattern, and that the W4 pattern in SST is independent of other natural variability such as Southern Annular Mode, Indian Ocean Dipole, or El Niño/Southern Oscillation.

While MHW have gathered considerable scientific attention, their cool cousins, marine cool spells (MCS) have attracted much less interest, even though MCS may also impact primary production, with consequent higher-trophic level impacts. Chiswell and O'Callaghan[16] showed that at least near the coast, cool spells can have a positive influence in production, likely due to associated upwelling. Off Western Australia, similar increased primary production during MCS was associated with recovery of benthic species that had been significantly impacted by previous MHW[17].

In this article, global reanalyses of sea surface temperature, sea-level atmospheric pressure, air-sea heat flux, and wind stress are used to investigate whether MHW/MCS in the southern hemisphere are becoming more frequent or more intense, and their relationship to global forcing.

In a warming ocean, if temperature anomalies were defined relative to a constant baseline value, there would be a trend towards more MHW and fewer MCS, so that eventually, the ocean would be in a continuous state of MHW. It is likely that the mechanisms forcing MHW/MCS are not the same as those driving long-term warming, and so it is important to separate MHW/MCS from any global warming. Thus, temperature anomalies were defined with respect to the 1982–2020 linear trend. Following Hobday et al.[18], SST anomalies were normalised by their 90th percentile to take into account local variability. There are two regions in the southern hemisphere where the SST anomaly exceeded three times its 90th percentile for more than 25 days between 1982 and 2020—the eastern Tasman Sea and the central South Pacific Ocean (Fig. 1). Of these, the Tasman Sea is of most interest because of its proximity to land, and because it spans the productive Subtropical Front[19] and is more likely to have societal impact. Consequently, this article focusses on this region.

Here I show that when defined by SST anomalies exceeding the 90th percentile for at least 5 days, 22 MHW and 21 MCS occurred in the Tasman Sea between 1982 and 2020. These events co-occur with corresponding events in the Atlantic, Indian, and eastern-Pacific Oceans, in a W4 structure. Canonical MHW and MCS constructed from these events show these events are likely driven by a stalling of a W4 atmospheric wave. The implications of this research are that while MHW/MCS in the Tasman Sea are a response to local wind field anomalies, they are driven by global atmospheric patterns, and ultimately predicting MHW/MCS requires understanding of, and an ability to predict, the W4 atmospheric wave.

## Results

**1982–2020 SST trend**. The southern hemisphere SST 1982–2020 trend (Fig. 1a) is consistent with trends for similar periods published elsewhere[20,21]. Highest warming, greater than 0.3 °C decade$^{-1}$, occurred in the western Tasman Sea and in the central South Pacific Ocean, and to a lesser extent, east of South America and in the Indian Ocean. Cooling occurred in the east Pacific Ocean near 20°S, and in the Southern Ocean, with maximum cooling about 0.3 °C decade$^{-1}$ south-east of South America.

Once the trend and annual cycles are removed, the 90th percentile of SST anomalies from 1982–2020 was typically 0.75 °C to 1 °C over most of the Southern Hemisphere, with highest values along the Pacific Ocean equator, along a zonal band south of Africa associated with the Agulhas Current retroflection[22], and east of South America associated with the Brazil-Falklands/Malvinas Confluence[23] (Fig. 1b). The number of days when the SST anomaly normalised by its 90th percentile, $SST_A$, exceeded 3.0 during 1982–2010 (i.e., when SST exceeded the category 3 MHW criterion[18]), shows high values near the equator (presumably reflecting El Niño events), in the central southern Pacific Ocean, and in the eastern Tasman Sea, where $SST_A$ exceeded 3 for more than 25 days (Fig. 1c).

**Tasman Sea MHW and MCS**. Based on Fig. 1c, and given the existing usage of the Tasman Box, the normalised SST anomaly averaged over this box, $SST_{Tas}$, was computed as an index of MHW activity in the Tasman Sea. This index has a mean value of 0.0, a standard deviation of 0.54, a skewness of 0.27, and a kurtosis of 3.8, indicating that its distribution is slightly skewed positive and is slightly fat-tailed. The index ranged from −2.1 to 2.7 (Fig. 2), with 22 warm events where $SST_{Tas}$ exceeded 1.0 for at

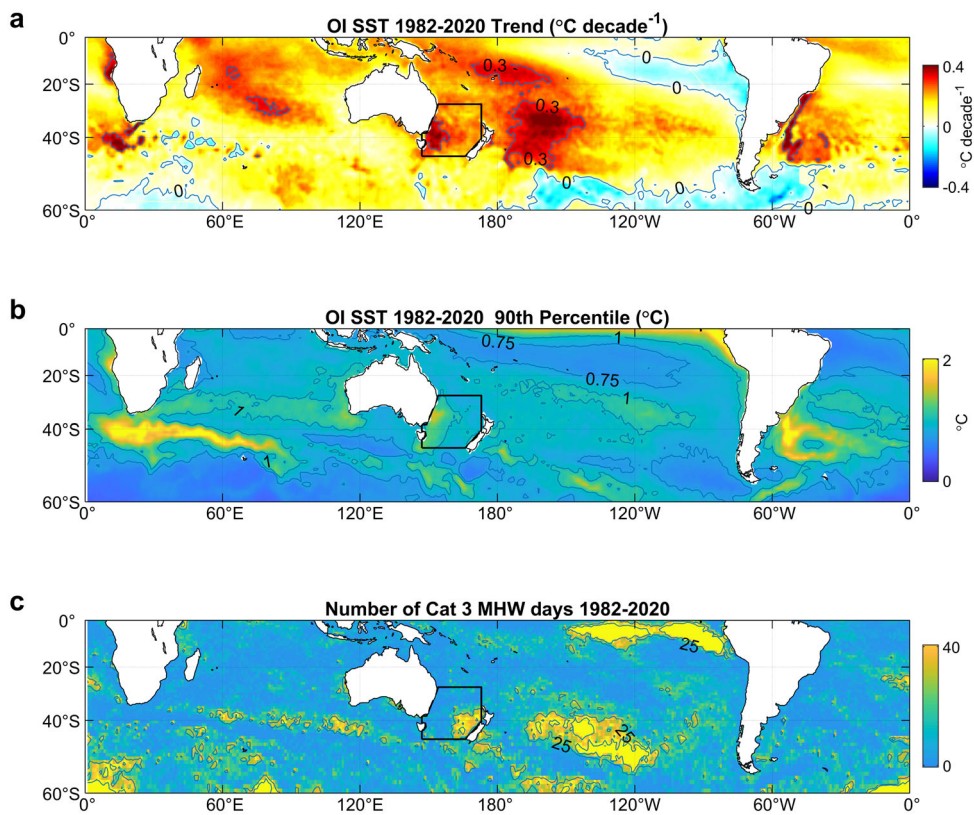

**Fig. 1 Sea surface temperature (SST) trend, 90th percentile, and number of warm days. a** 1982–2020 trend in SST computed from the Reynolds OISST reanalysis. **b** 90th percentile in SST anomalies once the annual cycle and linear trend are removed. **c** Number of days between 1982 and 2020 where the SST anomaly exceeded 3 times the 90th percentile. Black lines show the Tasman Box used to compute the Tasman Sea averaged SST anomaly ($SST_{TAS}$).

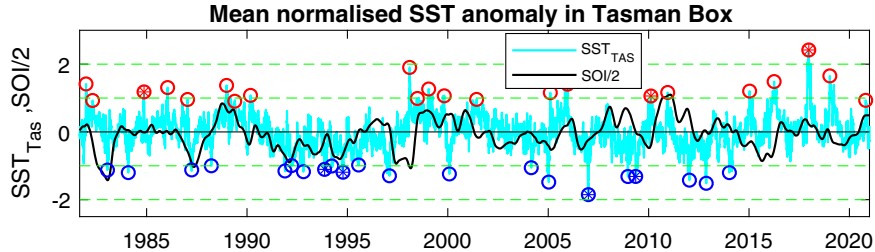

**Fig. 2 Tasman Sea averaged sea surface temperature anomaly ($SST_{TAS}$) and Southern Oscillation Index (SOI).** $SST_{TAS}$ (i.e., with trend and annual cycle removed, and normalised by the 90th percentile, cyan line) was computed over the box shown in Fig. 1. Circles show warm and cool events when $SST_{TAS}$ was greater than 1 or less than −1 for at least 5 d. Filled circles indicate the events shown in Fig. 3. Also shown is the SOI smoothed with a 6-month window and divided by 2 (black line).

least 5 d, and 21 cool events where $SST_{Tas}$ was less than −1.0 for at least 5 d.

Many of these warm events correspond to documented MHW, but because the index is averaged over the Tasman Box, the intensities of individual warm events in $SST_{Tas}$ do not necessarily correspond to peak intensities of individual MHW. For example, the extreme MHW in 2015/16 seen mostly in the western Tasman Sea[24] appears only as a moderate event in $SST_{Tas}$. However, the three most extreme warm events in the index correspond to well documented MHW—in February 1998[13], December 2017[4], and January 2019[16], with a combined average peak $SST_{Tas}$ of 2.23. The three strongest cool events were in January 2005, January 2007 and November 2012, with a combined average peak $SST_{Tas}$ of −1.8.

Linear regressions show no statistically significant change in intensity of warm events, ($r^2 = 0.12$, $p = 0.1$), although there is a

decade, $r^2 = 0.38$, $p = 0.003$) driven by the cluster of weak cool events in the mid-1990s followed by strong cool events in the 2010s. There is little evidence of any change in frequency of warm events, with 12 warm events in the first half of the record compared with 10 warm events in the second half. Cool events show a decline from 13 events in the first half to 8 events in the second half of the record. There is a significant trend towards longer-duration warm events (slope = 7 d decade$^{-1}$, $r^2 = 0.3$, $p = 0.01$), with the mean duration increasing from 8 d during the 1980s to 26 d during the 2010s, but there is no significant change in the duration of cool events (mean value = 12 d, $p = 0.18$).

Prior to 2005, $SST_{TAS}$ is positively correlated with the Southern Oscillation Index (SOI) showing a maximum of 0.36 (2.1 times is significance value, $\sigma_{95}$) at a lag of 90 days (SOI leading), with all but one cool event occurring when the SOI was negative and all but two warm events occurring when the SOI was near zero or positive. However, after 2005, $SST_{TAS}$ is negatively correlated with

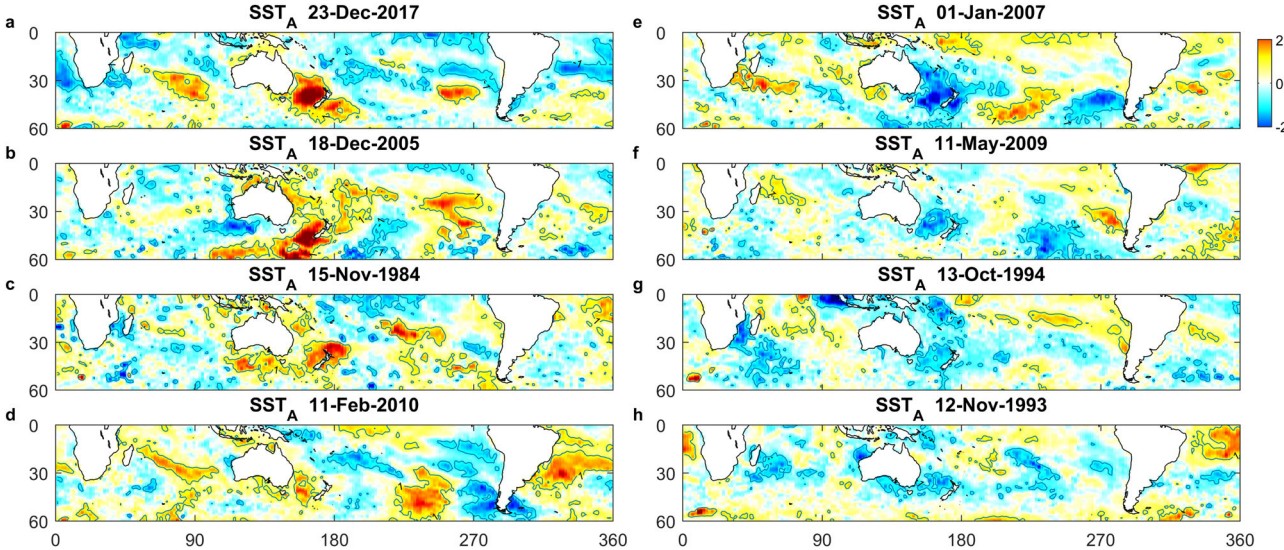

**Fig. 3 Representiative warm and cold events. a–d** Representiative warm events (defined when Tasman Sea averaged sea surface temperature anomaly ($SST_{TAS}$) was greater than 1, see Fig. 2). **e–h** Representiative cool events ($SST_{TAS} < -1$). **a** and **e** 23 December 2017 and 1 January 2007 show the strongest warm and cool events, respectively.

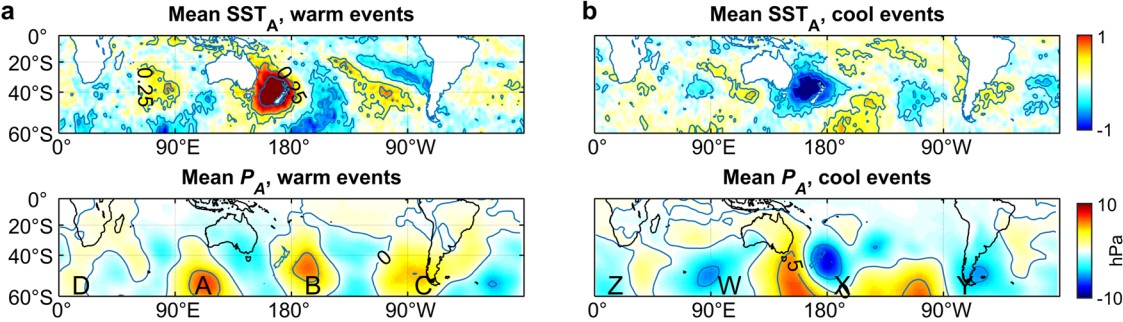

**Fig. 4 Mean sea surface temperature (SST) and atmospheric pressure anomalies during warm and cool events. a** Mean SST anomaly ($SST_A$) of all 22 warm events (indicated by red circles in Fig. 2), and mean atmospheric pressure anomaly ($P_A$) for the same events. **b** Mean $SST_A$ and $P_A$ for all 21 cool events (indicated by blue circles in Fig. 2). A to D and X to Z indicate highs and lows discussed in the text.

SOI, showing a peak of $-0.54$ ($3 \times \sigma_{95}$) at a lag of 270 days (SOI leading).

**Southern hemisphere MHW and MCS.** Space precludes showing all 43 warm and cool events, however, to illustrate the variability in their spatial structure $SST_A$ is presented for the peak of four representative warm and cool events (every 5th event when sorted by $SST_{TAS}$, Fig. 3). While there is considerable variability from event to event, the warm events often show regions in addition to the Tasman Sea, where $SST_A$ exceeds 2—in the Indian, south Pacific, and Atlantic Oceans. Similarly, cool events often show regions where $SST_A < -1$ off the west coast of South America, with weaker cool anomalies in the Atlantic Ocean.

The average of $SST_A$ at the peak of all 22 warm events (Fig. 4) has a maximum value in the Tasman Sea, as would be expected, where mean $SST_A > 1.0$, but in addition, there are regions in the Atlantic and Pacific Oceans where mean $SST_A$ exceeds 0.5, and a lessor region in the Atlantic Ocean where mean $SST_A$ exceeds 0.25. The average atmospheric pressure anomaly ($P_A$) for all 22 warm events shows a clear W4 structure with three strong highs (mean $P_A > 5$ hPa, labelled A to C), and a 4th, weaker high south of Africa (D). Each pressure high is centred south-east of a respective high in $SST_A$.

Mean $SST_A$ averaged over all 21 cool events is similar to mean $SST_A$ of warm events, but with opposite sign, so that every region

of positive $SST_A$ in the mean warm event has a corresponding region of negative $SST_A$. Mean $P_A$ for the cool events shows a W4 structure, but shifted about 45° in longitude compared to the mean of warm events. Three deep lows ($P_A < -5$ hPa, labelled W to Y) occur in about the same locations as the strong highs in the mean warm event, with a shallower low (Z) south of Africa.

**Canonical MHW and MCS.** Figure 4 alone is strong evidence that on average, MHW/MCS in the Tasman Sea co-occur with corresponding warm and cold anomalies in the Indian, Pacific and Atlantic Oceans, and that the likely drivers are a wavenumber-4 anomalies in atmospheric pressure. Based on this I constructed canonical MHW and MCS from averages of all 22 warm events and all 21 cool events. Figure 5 illustrates the $SST_A$ and $P_A$ progression of these canonical MHW and MCS from 45 d prior to the peak to the peak in $SST_A$ (the peak $SST_A$ fields are those shown in Fig. 4).

In the canonical MHW, 45 d prior to the peak of the event, there is little evidence of positive $SST_A$ in the Indian Ocean or Tasman Sea, although an area of positive $SST_A$ appears in the subtropics east of New Zealand (centred at 35°S, 145°W). At 30 d before peak, positive $SST_A$ anomalies begin to appear in the Tasman Sea and Indian Ocean. Over the next month, these positive anomalies intensify to reach maximum intensity and size at peak event.

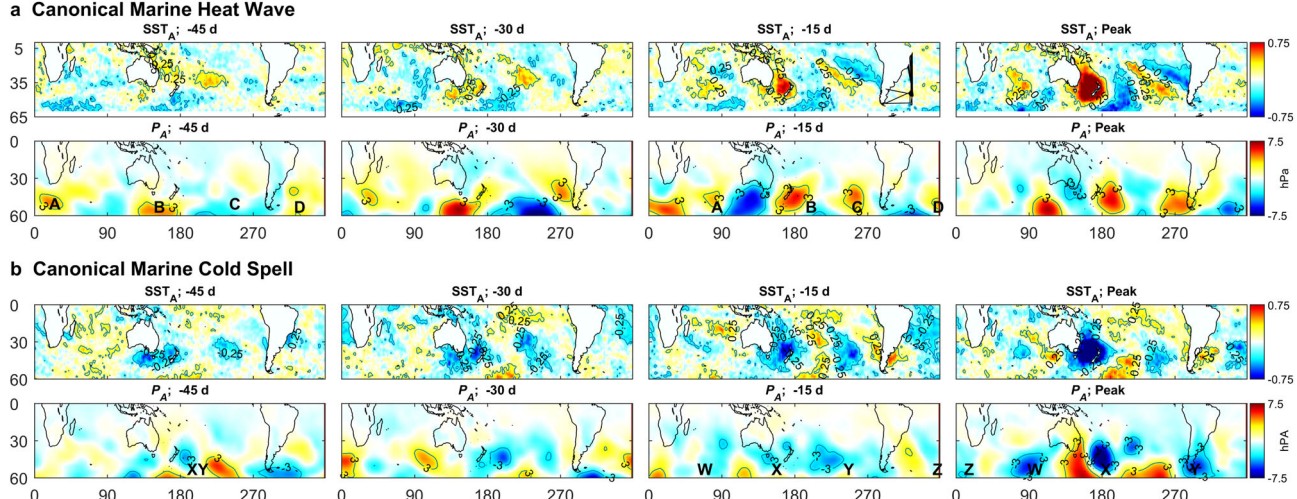

**Fig. 5 Progression of canonical marine heat wave and marine cool spell. a** Canonical marine heat wave (MHW) showing sea surface temperature anomaly ($SST_A$) and atmospheric pressure anomaly ($P_A$) from 45 d prior to peak event in Tasman Sea, to peak event. A to D indicate highs in $P_A$ discussed in the text. **b** Corresponding values for canonical marine cool spell (MCS). XY and W to Z indicate lows in $P_A$ discussed in the text.

Forty-five days prior to the peak in canonical MHW, atmospheric pressure anomaly shows the three highs ($P_A > 3$ hPa) south of Africa (A), Australia (B), and in the Atlantic Ocean (D), with a weaker high (>2 hPa) west of South America (C). Over the next 45 d, the high south of Africa (A) propagates eastwards while the Australian high (B) moves eastwards more slowly, so that an intense low develops between them. By 15 d prior to the peak, the Australian high sits over and to the east of New Zealand, and remains stationary until the peak, then it dissipates.

The canonical MCS develops somewhat similarly, but with opposite sign, so that the highs (A, B, C) are replaced by lows (W, X, Y). The correspondence is not exact, with the main difference being that MCS appear to develop earlier than MHW with cool SST anomalies appearing around New Zealand by 45 d prior to peak. It is also not so easy to track the progression of atmospheric pressure lows across the globe.

The temporal development of the canonical MHW/MCS is perhaps better illustrated by Hovmöller diagrams of $SST_A$ and $P_A$ along with air-sea heat flux and wind stress anomalies ($Q_A$ and $\tau_A$) at 45°S (Fig. 6). In the canonical MHW, SST anomalies of 0.25 begin to appear about 40 d prior to the peak, and last to about 50 d post peak. At the beginning of the record, $P_A$ shows clear evidence of eastward propagation at ~5° d$^{-1}$, with a mean period of about 20 d. At least two coherent anomalies in $P_A$ can be tracked across the entire globe (indicated by the red and black dashed lines). About 40 d prior to the peak, however, this coherent propagation breaks down, and the highs (A to D) stall and intensify, and then propagate eastward at a slower rate of about 1–2° d$^{-1}$. About 20 d after peak, these highs dissipate, and the dominant 5° d$^{-1}$ eastward propagation returns. Similarly, $Q_A$ shows coherent 5° d$^{-1}$ eastward propagation until about 60 d prior to the peak, when anomalous lows (ocean heating) appear coincident with the atmospheric pressure highs. The strongest anomalies (−20 W m$^{-2}$) coincide with high B. Although not shown here, this signal is almost entirely due to latent heat. Wind stress does not show as strong a relationship with $P_A$, but there is a general tendency for negative $\tau_A$ to be associated with positive $P_A$, and the strongest lobe of the MHW in the Tasman Sea coincides with the strongest negative anomaly in wind stress.

The canonical MCS pattern is generally similar to that of the canonical MHW, but of opposite sign. The main differences are that Tasman Sea cooling ($SST_A < -.25$) appears much earlier than

MHW warming, and it is the Indian Ocean, rather than the Atlantic Ocean pole that shows weakest cooling. The earlier onset of MCS is due to the eastward propagation in $P_A$ stalling earlier than in the MHW. For example, the lows X and Y appear to develop from a low (XY) that stalls about 90 d prior to peak. Consequently, increased heat loss (positive $Q_A$) associated with the $SST_A$ cooling appears much earlier than corresponding values in the MHW. Positive wind stress anomaly coincides with negative $P_A$, and increased heat loss, especially along the track of low XY, indicating that winds are a strong driver of cooling in MCS. MCS tend to last longer than MHW, even though the lows in atmospheric pressure dissipate at about the same rate as the highs during MHW.

**Discussion**

In summary, when SST anomalies are defined with respect to a linear trend to take into account long-term change and normalised by their 90th percentile to take into account local variability, about 20 MHW and MCS each occurred in the Tasman Sea between 1982 and 2020, but these are not uniformly distributed in time. There is little evidence that MHW are increasing in either frequency or intensity in the Tasman Sea, although their duration may be increasing. MCS showed a significant increase in strength but appear to be decreasing in frequency. The main findings are that MHW/MCS in the Tasman Sea are part of a global phenomenon driven by a wavenumber-4 (W4) atmospheric forcing, and there is no clear relationship between MHW/MCS and SOI.

Removal of a long-term trend is predicated on the assumption that MHW or MCS are spells when SST is significantly warmer or cooler than expected, and that the expected value changes with time. This approach differs from researchers who do not remove a trend and conclude that MHW are increasing in intensity[1,2], but agrees with researchers who also remove the trend[25]. However, it has been suggested that the rate of global warming is increasing with time[26]. If so, a linear trend may not adequately represent the expected value, and there would be an influence on the derived statistics of MHW/MCS. For example, the conclusion that MHW increased in duration, while statistically significant, is driven by the fact that $SST_A$ tended to be positive over the last 5 years, leading to three longer-than normal warm events (as seen in Fig. 2). Similarly, the decrease in the frequency of MCS is driven by the lack of cool events over the last 7 years. Had a trend that allowed for accelerated heating been removed, these two

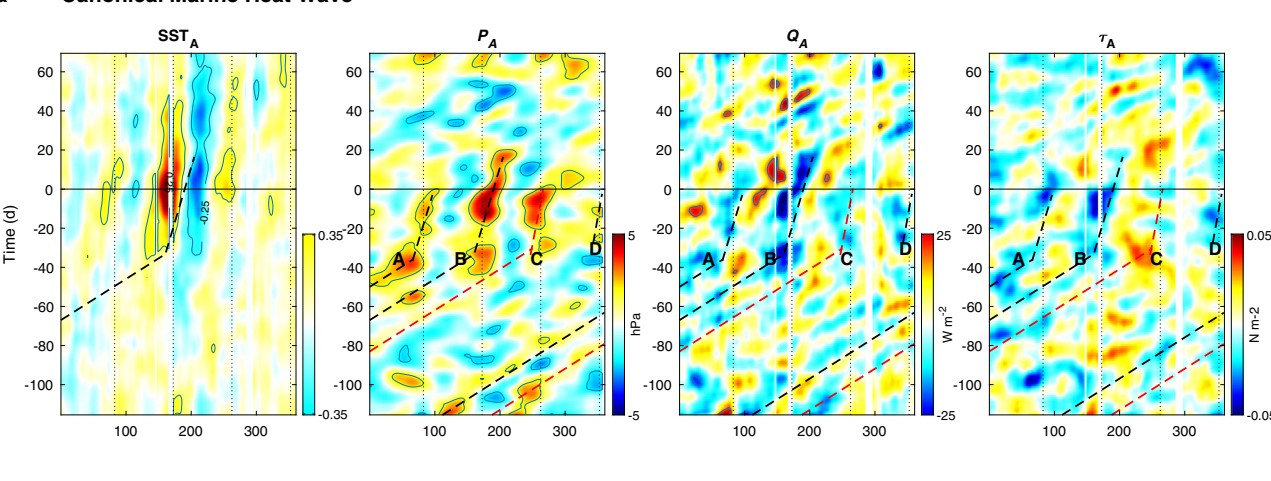

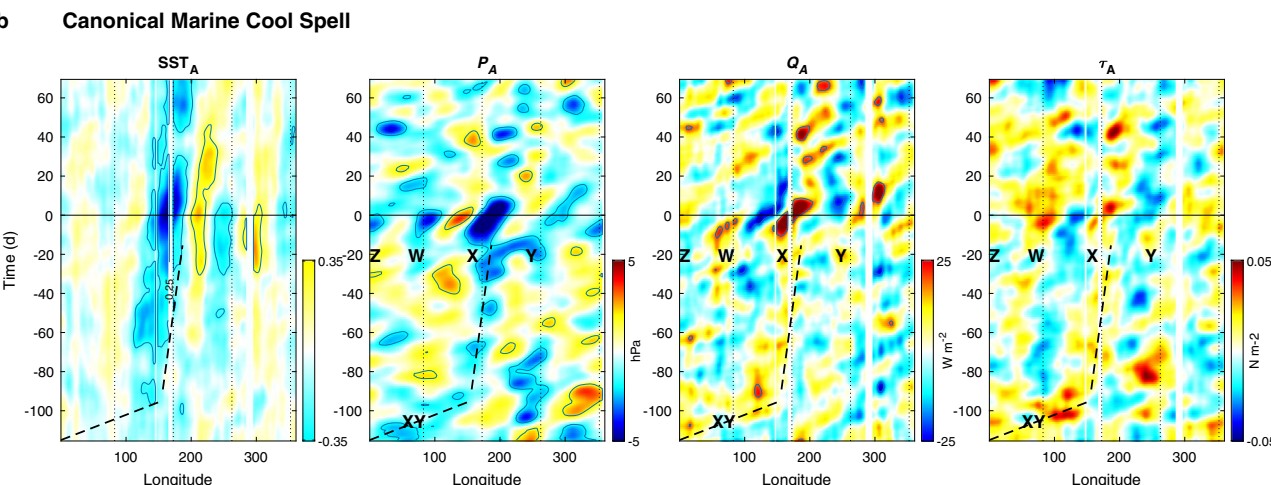

**Fig. 6 Hovmöller diagrams for canonical marine heat wave (MHW) and marine cool spell (MCS) at 45°S. a** Canonical MHW and **b** Canonincal MCS. Panels show zonally-averaged sea surface temperature anomaly ($SST_A$), atmospheric pressure anomaly ($P_A$), air-sea heat flux anomaly ($Q_A$), and wind stress anomaly ($\tau_A$) as a function of time. The peak event occurs at time = 0. Labels A to D and W to Z indicate pressure highs and lows shown in Figs. 4 and 5 and discussed in the text. Vertical dotted lines spaced 90° in longitude apart indicate the wavelenth of a W4 wave, with the middle line at the longitude of the eastern Tasman Sea. Blank areas indicate the land masses of Australia, New Zealand, and South America. Sloped dashed lines indicate a phase-speed of 5° d$^{-1}$. Steeper dashed lines indicate slower speeds ~1° d$^{-1}$ ascribed to highs (A to C) and lows (X and Y) in $P_A$. The heat convention is that negative values indicate heat entering the ocean.

conclusions would have been different. If the 10-year running mean had been removed instead of a linear trend, the increase in MHW duration would have been shorter and less significant (4 d decade$^{-1}$, $p = 0.04$, vs 7 d decade$^{-1}$, $p = 0.01$). This points not only to the importance of removing a trend, but also that the form of the trend impacts MHW/MCS statistics, and consequently there needs to be a common consensus on how to deal with long-term warming in MHW/MCS analyses.

Normalising by the 90$^{th}$ percentile is designed to account for regional variability in SST variance[18]. But this has a subtle effect in that any phenomena having globally similar amplitude would be scaled down in regions where other mechanisms lead to strong SST variability. This may in part explain why the otherwise quiescent eastern Tasman Sea appears to have so many MHW/MCS. If so, the canonical MHW/MCS somewhat mis-represent the SST response by emphasizing the eastern Tasman Sea. But by not normalising the atmospheric pressure, heat flux, or wind stress anomalies, this analysis presents regionally unbiased descriptions of the atmospheric forcing.

The peak canonical MHW and MCS fields alone (Fig. 4) are strong evidence that MHW/MCS are driven by a W4 atmospheric wave, but the main support comes from the Hovmöller diagrams.

There will be some error introduced by aligning events by the peak in $SST_{TAS}$, perhaps by several days for each event, and this presumably leads to some of the noise in these diagrams. Similarly, one would not expect all MHW/MCS to evolve at the same rate, and variations from one event to the next will also lead to noise, explaining why even when they are at their most coherent, it can sometimes be difficult to track individual events in either the ocean response or atmospheric forcing.

Overall, however, it is clear that the atmospheric pressure is dominated by W4 events that propagate eastwards, and that during MHW/MCS, this eastward propagation stalls for up to two months and so sets up the co-occurring MHW/MCS in the Tasman Sea, Indian, Pacific, and Atlantic Oceans.

During the canonical MHW, this slowdown in propagation sets up anomalous high pressure east of New Zealand which drives northerly/north-easterly- winds over the Tasman Sea that are weaker than usual, setting up a period of increased heat flux into the ocean. This canonical description broadly fits observations from the Tasman Sea of anomalously low air pressure[4] and southwards directed ocean currents[12] during MHW, although one would expect the relative roles of increased air-sea flux and advection to vary over the course of each MHW.

It has been conjectured that remote forcing by Rossby waves also contributes to Tasman Sea MHW by increasing poleward transport of the East Australian Current and its extension[13], but it is not clear how such a mechanism fits into the canonical MHW —there is no evidence of westward propagation associated with Rossby waves in the Hovmöller diagrams for 45°S (Fig. 6), or at any other latitude south of 25°S (not shown).

The canonical MCS is driven similarly, but the W4 atmospheric pressure events stall one-half wavelength zonally out of phase to the MHW event, setting up an anomalous low atmospheric pressure east of New Zealand driving northward advection of cooler surface water. In the canonical MCS, there is also anomalously stronger wind stress, adding a vertical mixing component to the cooling.

W4 structure of the SST variability is not confined to MHW/MCS, for example, Senapati et al.[15] report a stationary W4 pattern in SST set up in summer over the southern hemisphere, suggesting that it is set up by coupling between the atmosphere and ocean. Fauchereau, et al.[27] similarly found SST co-varied in the southern Atlantic and Indian oceans in a W4 spatial structure. Thus, this work suggests that the canonical MHW/MCS are manifestations of an austral W4 zonally propagating atmospheric wave that in some years stalls out and intensifies.

What drives this stalling and intensification is not obvious, but even with 40 years of data, it can be quite problematic to make inferences from correlations with atmospheric indices. Significant correlations with SOI changed sign (and lag) in about 2005. Had only the record prior to 2005 been available, one would have concluded a strong positive correlation between SOI and Tasman Sea temperature anomalies, but had only the record post 2005 been available, one would have come up with the opposite conclusion. As Holbrook et al.[9] note, relationships between MHW and climate modes are complex.

Finally, the canonical MHW/MHS derived in here are, by virtue of their definition, broad-scale events centred on the Tasman Sea. Individual MHW/MCS show considerable variability in both spatial extent and intensity (Fig. 3) so that while the large-scale atmospheric patterns appear to set up the MHW/MCS, the local response in the Tasman Sea is driven by shorter-scale variability in the forcing. Nevertheless, the results of this work suggest that the key to making accurate predictions of MHW/MCS globally lies in understanding the causes of, and being able to predict, the W4 atmospheric wave stalling and intensification.

## Methods

**Daily SST analysis**. Daily OISST reanalysis products[28] from 1 September 1981 to 31 December 2020 were obtained from NOAA. At each location, the annual cycles were first removed from the daily SST, then the 1982–2020 linear trend was removed. Following Hobday et al.[18], SST anomalies were then normalised by the local 90th percentiles to produce the normalised SST anomaly, $SST_A$. Daily reanalyses of atmospheric pressure, heat fluxes, and wind stress obtained from NCEP were treated similarly (annual cycles and trend removed), but not normalised to produce anomalies, $P_A$, $Q_A$, and $\tau_A$, respectively.

The Tasman Box has been defined by previous researchers as the region between 46°S and 28°S and between 147°E and 173°E[12]. The mean SST anomaly in the Tasman Box (i.e., normalised and weighted for grid area), $SST_{TAS}$, was computed and is shown in Fig. 2. Warm and cool events were then defined when $SST_{TAS}$ exceeded 1 or was less than −1, respectively for at least 5 days. Events were required to be separated by at least 90 days. This separation requirement was chosen so that the occasional case where $SST_{TAS}$ fluctuated around the limits over several weeks counted as one event to avoid biasing the mean statistics towards any particular year.

**Canonical MHW and MCS**. The 22 warm and 21 cool events thus identified were then averaged to form canonical MHW and MCS by aligning each event in time centred on the peak value of $SST_{TAS}$ and computing an evolving mean event extending two months either side of the peak. Corresponding timeseries of atmospheric pressure, air-sea heat flux, and wind stress anomalies were computed using the same alignments.

Hovmöller diagrams of $SST_A$, $P_A$, $Q_A$, and $\tau_A$ for 45°S were computed by averaging the respective quantities over a 5° latitude band and plotting against longitude.

**Statistical significance**. The significance of the slope of linear regressions was calculated following Santer et al.[29], where the trend is considered significantly different from zero if the ratio of the estimated trend and its standard error, $t_b$, falls above the Student's $t$-test value, given a confidence level and number of degrees of freedom. The confidence level was taken to be 95% and the number of degrees of freedom was computed as the record length minus 2, assuming every 15′th daily satellite estimate was independent (15 d was chosen as this is the lag when the $SST_{TAS}$ autocorrelation-squared is 0.5).

Lagged correlations and their standard errors were computed using the method described in Sciremammano[30]. Correlations are considered significant at the 95% level when they exceed one standard error.

## Data availability

NOAA 1/4° daily Optimum Interpolation Sea Surface Temperature (OISST) are available from https://www.ncdc.noaa.gov/oisst. NCEP daily reanalyses of atmospheric pressure, heat fluxes and wind stress were obtained from http://www.esrl.noaa.gov/psd/data/gridded/data.ncep.reanalysis.surfaceflux.html. The Southern Oscillation Index (SOI) index was obtained from the Australian Bureau of Meteorology website http://www.bom.gov.au/climate. Derived data, including SST anomalies and the Tasman Sea index, generated in this study have been deposited in the Zenodo database under accession code doi:10.5281/zenodo.5096921.

## Code availability

Matlab code used in this analysis has been deposited in the Zenodo database under accession code https://doi.org/10.5281/zenodo.5096921.

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

## Acknowledgements

The author thanks NOAA for making the OISST and NCEP reanalysis products freely available. This work was supported by the New Zealand Ministry of Business, Innovation and Employment through grants to NIWA's Climate, and Coasts and Oceans pro-grammes and their predecessors.

## Author contributions

S.M.C. conceived the study, performed the analyses, and wrote the manuscript.

## Competing interests

The authors declare no competing interests.
