## [Peer Review File · Nature Communications]

REVIEWER COMMENTS

Reviewer #1 (Remarks to the Author):

This paper details an analysis that shows a significant atmospheric driver pattern for both Marine heatwaves and to a lesser extent cool spells in the Tasman Sea. There is much interest in moving from documenting marine heatwave episodes and trends and further investigate drivers which will assist models in predicting heatwaves in the future.

When reading the manuscript, the term wavenumber is front and centre in the title and abstract. This word needs further explanation, and I wouldn't assume readers will understand this (especially when looking at the title only). No further explanation is provided in the text, only references, but this needs to be explained herein for readers.

The MHW assessment has been done on a detrended SST. I can understand why this has occurred, by removing warming trends as a driver and focusing on other drivers. However the abstract states that there has been no increase in frequency or duration, which is a conclusion that needs to be flanked with qualifiers, especially in the abstract. Line 244/245 mentions that removing the trend 'perhaps' force this conclusion, but I would say it absolutely does. I think the conclusion is valid in terms of the atmospheric drivers, but this needs to be explicit throughout when mentioned that trends are referring to MHW subjected to non-warming trends.

120: I had to read this sentence a few times, it sounds like you are saying when the number of days exceed 3 (when they need to be 5 for MHW). Please say 'exceeded category 3' both times 'exceeded 3' is stated.

202: The assumption that XY splits is not supported by these figures, unless there are additional days that show this. The atmospheric progression is far less obvious, and needs further clarification to the conclusion drawn here.

245: works, not workers

252: organisms that that are killed by 10 d of warming might have survived 4 decades ago but would not survive today.
Why?

Figure Captions, please ensure TAS is in subscript in SSTAS

Figure 6: Australia misspelt

Reviewer #2 (Remarks to the Author):

General Comments

The author presents a historical analysis of marine heat waves (MHW) and cold spells (MCS) in the Tasman Sea, an area that has received considerable attention in the literature for extreme temperature events. A composite analysis of events observed over the past ~40 years reveals longitudinal structure in Southern Hemisphere patterns of SST and atmospheric pressure anomalies, which are associated with the previously described atmosphere wavenumber 4. The analysis is straightforward and offers a clear relationship between atmospheric anomalies and the ocean response in this region, though the writing and figures need to be improved.

Specific Comments

Title: Suggest adding "in the Tasman Sea" to the end of the title. While the author also shows anomalies in other regions associated with wavenumber 4, they are on average not strong enough to be MHW/MCS

L23-25: I'm not sure about the justification for calling the Tasman Sea a "hotspot". As far as I can tell, it's not based on the Tasman Sea having particularly long or strong heatwaves, it's based on the somewhat arbitrary "number of days above 3 times the 90th percentile SST anomaly".

L40: This statement is misleading. Because of the way Oliver et al. define MHWs, their "increasing MHWs" is actually just the warming trend, not something separate from it. There is good text at L93-98, and that context should be provided up front.

L60: Saying that Bond et al. supports the earlier papers is circular since it's one of the studies they base their conclusions on

Intro: Somewhere it would be good to expand on the meaning and significance of the atmospheric wavenumber 4

L84-85: presumably they can impact much more than just primary production – e.g., marine species populations, physiology, distributions...

L86-87: In this case it would be the input of upwelled nutrients influencing production, not the cold spell itself

Figures 1,3,5,6: It's best practice to avoid the rainbow colormap. It would also be nice to have some variation in colormaps for panels/figures showing different quantities. Last, for anomalies (e.g., Fig. 3) a colormap with white at zero (e.g., blue-white-red) is easy to interpret.

Figure 1,4,5,6: Need scales for what the colors represent.

Figure 1: Panel (c): Is there also a 5 day threshold applied here, or is this the total number of days exceeding the threshold?

L109-113: Both cooling and warming are described southeast of South America – could revise language to clarify.

L119-124: It would be worth mentioning the source of this pattern. By my understanding, it must reflect the degree to which the SSTa distributions are non-normal (i.e., skewed or fat-tailed).

L144-145: How did you decide there is no decrease in cool events? (13->8 is a ~40% decline)

L150-151: Do you think this increase in duration is a robust finding? If so it would be worth some analysis (or at least discussion) of why there might be an increase.

L153-159: Reference Figure 2 here. Also could provide the SSTAS-SOI correlations for different periods instead of saying the "appear" to be positively/negatively correlated.

L160-162: Here and elsewhere - rather than opening a sentence/paragraph talking about figures (e.g., Figure X shows...), state the point and reference the figure to back it up. I think Fig. 3 could be eliminated (Fig. 4 makes this point more clearly)

L169: missing "are"

L201: The dissipation is not shown – it would be interesting to see the evolution after the peak too

L209-226: The faster eastward propagation is evident to me in Qa for MHW and MCS, and (to a lesser extent) Pa for MHW. For all the other variables, I don't see propagation as suggested by the dashed lines.

L237-238: It's still unclear to me what it means to say it's a major region for MHWs. The basis for this seems to be "days above 3x the 90th percentile", which again is really just a measure of skewness in the SST anomaly distribution.

L241-245: I appreciate this approach, which is the correct one since waves are by definition variability relative to the mean state. This is an ongoing discussion in the field (e.g., Jacox et al., Nature, 2019), and this study is a good example of one reason to remove trends (so that mechanisms can be separated from long-term warming). I disagree with the statement at L243-245 though; after detrending there can still be changes in intensity/frequency, which would come from changes in the SST anomaly distribution (e.g., increasing/decreasing variance).

L247-253: I think this paragraph should be cut – it doesn't really fit and doesn't draw on the results at all.

L273-283: This explanation isn't clear to me, particularly because the anomalous winds are not apparent. From the results presented here, the main driver of MHW/MCS appears to be the surface heat flux anomalies (which the author says are dominated by the latent heat flux). Please clarify the explanation.

Discussion: Somewhere it would be good to discuss the implications of the findings. e.g., might there be predictability associated with the atmospheric pattern.

L307-308: Were the 90th percentile anomalies the same for the whole year or were they seasonally dependent (as in Hobday et al)?

L313-316: Inconsistency here: is SSTAS the area averaged SST (as stated in 313-315) or the area averaged normalized SST anomaly (as suggested in 315-316)?

L316-317: What is the justification for a 90 day separation? I haven't seen anything like that before and I can't think of a reason for it. And if there are events within 90 days of each other how do you choose which to keep?

L326-327: Please provide a brief description of the methodology that does not require reading Santer et al.

Mike Jacox
NOAA

REVIEWER COMMENTS

Reviewer #1 (Remarks to the Author):

This paper details an analysis that shows a significant atmospheric driver pattern for both Marine heatwaves and to a lesser extent cool spells in the Tasman Sea. There is much interest in moving from documenting marine heatwave episodes and trends and further investigate drivers which will assist models in predicting heatwaves in the future.

When reading the manuscript, the term wavenumber is front and centre in the title and abstract. This word needs further explanation, and I wouldn't assume readers will understand this (especially when looking at the title only). No further explanation is provided in the text, only references, but this needs to be explained herein for readers.

I have now included a brief description of wave number-4 mode when first introduced 'where a dominant mode in SST anomalies showed four highs spaced about 90° of longitude apart (south of Australia, central Pacific, western Atlantic and western Indian Oceans) interspersed with four lows'

The MHW assessment has been done on a detrended SST. I can understand why this has occurred, by removing warming trends as a driver and focusing on other drivers. However the abstract states that there has been no increase in frequency or duration, which is a conclusion that needs to be flanked with qualifiers, especially in the abstract. Line 244/245 mentions that removing the trend 'perhaps' force this conclusion, but I would say it absolutely does. I think the conclusion is valid in terms of the atmospheric drivers, but this needs to be explicit throughout when mentioned that trends are referring to MHW subjected to non-warming trends.

The journal requirement of a 150-word limit means that I had to completely rewrite the abstract, and unfortunately did not have room to add these qualifiers.

Reviewer 2 points out that it is possible that removing the trend does not force the conclusion that there is no increase in frequency or intensity of MHW/MCS – (for example, longer or stronger MHW could be compensated for by longer or stronger MCS), so this sentence has been deleted.

There is now more discussion on the trend removal and its implications in the Discussion.

120: I had to read this sentence a few times, it sounds like you are saying when the number of days exceed 3 (when they need to be 5 for MHW). Please say 'exceeded category 3' both times 'exceeded 3' is stated.

Since SSTA is non-dimensional, it would not be technically correct to say 'when SSTA exceeded category 3'. I have changed the sentence to "The number of days when the SST anomaly normalised by this 90th percentile, SST_A , exceeded 3.0 during 1982-2010 (i.e. when SST exceeded the category 3 MHW criterion, Hobday et al., 2018) ..."

202: The assumption that XY splits is not supported by these figures, unless there are additional days

that show this. The atmospheric progression is far less obvious, and needs further clarification to the conclusion drawn here.

The conclusion that XY split is better made from the Hovmuller diagram (Figure 6). The pressure panel shows a low appearing in the west at about -100 days. This low tracks east but then stalls near the longitude of NZ at about -80 d. This low then remains more or less stationary until about -40 days when it appears to split into X and Y. The QA and tauA panels show similar features (with positive QA and tauA corresponding to negative PA).

I have changed the text to make this conclusion based on an interpretation of the Hovmuller diagrams (Figure 6) rather than the sequence of events.

245: works, not workers

'workers' has been changed to 'researchers'.

252: organisms that that are killed by 10 d of warming might have survived 4 decades ago but would not survive today.

Why?

This paragraph was meant to illustrate the biological impacts of longer MHW now than 40 years ago. Reviewer 2 suggested eliminating this paragraph, and I have done so.

Figure Captions, please ensure TAS is in subscript in SSTAS

This was due to a bug in MS Word cross-referencing where subscripts in the figure list are lost when captions are copied to the figures. Hopefully now corrected.

Figure 6: Australia misspelt

Corrected

Reviewer #2 (Remarks to the Author):

General Comments

The author presents a historical analysis of marine heat waves (MHW) and cold spells (MCS) in the Tasman Sea, an area that has received considerable attention in the literature for extreme temperature events. A composite analysis of events observed over the past ~40 years reveals longitudinal structure in Southern Hemisphere patterns of SST and atmospheric pressure anomalies, which are associated with the previously described atmosphere wavenumber 4. The analysis is straightforward and offers a clear relationship between atmospheric anomalies and the ocean response in this region, though the writing and figures need to be improved.

Specific Comments

Title: Suggest adding “in the Tasman Sea” to the end of the title. While the author also shows anomalies in other regions associated with wavenumber 4, they are on average not strong enough to be MHW/MCS

I have changed the title to “Atmospheric Wavenumber-4 driven South Pacific Marine Heat Waves and Marine Cool Spells”

L23-25: I’m not sure about the justification for calling the Tasman Sea a “hotspot”. As far as I can tell, it’s not based on the Tasman Sea having particularly long or strong heatwaves, it’s based on the somewhat arbitrary “number of days above 3 times the 90th percentile SST anomaly”.

I have eliminated the term ‘hotspot’, and now make the case for concentrating on the Tasman Sea since of the two regions in the southern hemisphere where there are a large number of days when the anomaly exceeds the 90th percentile, it (the Tasman Sea) is likely to have more biological and societal impact.

L40: This statement is misleading. Because of the way Oliver et al. define MHWs, their “increasing MHWs” is actually just the warming trend, not something separate from it. There is good text at L93-98, and that context should be provided up front.

I did not want to go into much detail about this point in the Introduction since it is dealt with more extensively in the Discussion, and did not want the paper to be repetitive.

I have added the following in parenthesis after quoting Oliver et al “...(although it has been pointed out that such suggestions may not take into account long-term warming (Jacox, 2109)).”

L60: Saying that Bond et al. supports the earlier papers is circular since it’s one of the studies they base their conclusions on

I have removed the text ‘Supporting this,’

Intro: Somewhere it would be good to expand on the meaning and significance of the atmospheric wavenumber 4

I have added a line to the Introduction saying that the implications of this research are that predicting MHW/MCS in the Tasman Sea require understanding of, and an ability to predict the W4 atmospheric wave.

There is a new paragraph at the end of the Discussion expanding on this (see response to a similar comment below).

L84-85: presumably they can impact much more than just primary production – e.g., marine species populations, physiology, distributions...

I have changed these lines to ‘MCS may also impact primary production, with consequent higher-trophic level impacts. Chiswell and O’Callaghan (2021) showed that at least near the coast, cool spells can have a positive influence in production, likely due to associated upwelling. Off Western Australia, similar increased primary production during MCSs was associated with recovery of benthic species that had been significantly impacted by previous MHWs(Feng et al., 2021).

L86-87: In this case it would be the input of upwelled nutrients influencing production, not the cold spell itself

Agreed. At the coast, the MCS were associated with upwelling favourable winds. Presumably in the open ocean MCS could be caused by a number of factors, such as increased convective overturn, increased wind stress or increased equatorially-directed transport. How these factors impact primary production would depend on the particular biome, i.e., how the production is determined by the physics – whether, for example, increased mixing introduces more nutrients into the upper layers leading to increased production, or mixes plankton below the photic zone leading to reduced production.

Rather than get into this discussion, I have just stated ‘but it is not so clear if they also have an impact in the open ocean’ in the paragraph above.

Figures 1,3,5,6: It’s best practice to avoid the rainbow colormap. It would also be nice to have some variation in colormaps for panels/figures showing different quantities. Last, for anomalies (e.g., Fig. 3) a colormap with white at zero (e.g., blue-white-red) is easy to interpret.

I have replaced the rainbow colour map with one that has white at zero for quantities showing both positive and negative values (e.g. anomalies). Other quantities now are shown with a parula colormap.

Figure 1,4,5,6: Need scales for what the colors represent.

I originally omitted scales to reduce the complexity of the figures (thinking that having contours would provide a scale). I have now added scales to all figures.

Figure 1: Panel (c): Is there also a 5 day threshold applied here, or is this the total number of days exceeding the threshold?

This is just the number of days when SST anomalies were warmer than 3 times the 90th percentile. There was no 5-day threshold.

L109-113: Both cooling and warming are described southeast of South America – could revise language to clarify.

I now say “Highest warming , greater than $0.3^{\circ}\text{C decade}^{-1}$, occurred in the western Tasman Sea and in the central South Pacific Ocean, and to a lesser extent east of South America and in the Indian Ocean. Cooling occurred in the east Pacific Ocean near 20°S , and in the Southern Ocean, with maximum cooling about $0.3^{\circ}\text{C decade}^{-1}$ south-east of South America

L119-124: It would be worth mentioning the source of this pattern. By my understanding, it must reflect the degree to which the SSTa distributions are non-normal (i.e., skewed or fat-tailed).

Indeed - If normally-distributed, SSTa would exceed three times the 90th percentile only about 0.0062 % of the time – ie about 1 day between 1982 and 2020, whereas we get upwards of 25 days over that time period in places. However, there are also about the same number of cool events, so the distribution is fat-tailed .

Rather than discuss the distributions of SSTa, I briefly discuss the distribution of the Tasman Sea index SST_{TAS} in the next paragraph by including the sentence “This index has a mean value of 0.0, a standard deviation of 0.54, a skewness of 0.27 and a kurtosis of 3.8, indicating that its distribution is slightly skewed positive and slightly fat-tailed.”

L144-145: How did you decide there is no decrease in cool events? (13->8 is a ~40% decline)

The problem with trying to decide if there has been a change in frequency of events is that there have been multi-year periods in the past with no or few events. For example, there were no MHW over most of the 1990’s, and a 7-year period in the 2000’s with only one MCS. The lack of MCS over the past 6/7 years may just be a slightly longer one of these multi-year periods with no events.

I now include discussion of this result in the second paragraph of the Discussion which now discusses the impact of removing a linear trend on both the findings of a decrease in cool event frequency and increase in duration of warm events.

L150-151: Do you think this increase in duration is a robust finding? If so it would be worth some analysis (or at least discussion) of why there might be an increase.

The increase in duration comes out as statistically significant. But as I think the reviewer will agree, statistically significant results do not necessarily mean a robust finding. In particular,

this result is driven by the fact that SSTA tended to be positive over the last 5 or so years, leading to three longer-lasting warm events (as seen in Figure 2). This is a result of using a linear trend as a baseline that does not capture the higher than usual warming over the last 5 years.

Had I used a different baseline, this finding might have been different.

I now include discussion of this result in the second paragraph of the Discussion - see response to comment above.

L153-159: Reference Figure 2 here. Also could provide the SSTAS-SOI correlations for different periods instead of saying the “appear” to be positively/negatively correlated.

I now present the SSTAS-SOI lagged-correlations for pre- and post-2005. This point is reinforced by inclusion of a second panel in Figure 2 showing SSTAS and rescaled SOI for each half of the record separately, which is further discussed near the end of the Discussion.

L160-162: Here and elsewhere - rather than opening a sentence/paragraph talking about figures (e.g., Figure X shows...), state the point and reference the figure to back it up. I think Fig. 3 could be eliminated (Fig. 4 makes this point more clearly)

I agree that Figure 4 makes the point more clearly, but am loathe to remove Figure 3 because it gives a sense of the interannual variability in the extent and strength of the various MHW.

I have tried to minimise opening a paragraph or sentence with “Figure X shows ...”, but in a couple of cases, alternative wording seemed cumbersome and unclear.

L169: missing “are”

These two sentences have been rephrased.

L201: The dissipation is not shown – it would be interesting to see the evolution after the peak too

I tried adding another panel, but then the panels become so small and the figure overcrowded that the relevant features become difficult to see, especially when printed on a journal page.

The dissipation can be inferred from Figure 6, which shows that MCS tend to last longer than MHW. A sentence to this effect is now added in discussion of Figure 6.

L209-226: The faster eastward propagation is evident to me in Q_a for MHW and MCS, and (to a lesser extent) P_a for MHW. For all the other variables, I don't see propagation as suggested by the dashed lines.

The dashed lines were meant to be guides to help the reader compare the various panels in the figure, and were not meant to suggest that there was eastward propagation in all cases (for example SSTA shows little eastward propagation).

The text relating to the Hovmuller diagrams has been changed, and the dashed lines have been changed to help the discussion.

L237-238: It's still unclear to me what it means to say it's a major region for MHWs. The basis for this seems to be "days above 3x the 90th percentile", which again is really just a measure of skewness in the SST anomaly distribution.

This paragraph (summarising the findings) has been completely rewritten and rather than saying the Tasman Sea is a major region for MHW, I say that when defined by Cat 3 definition, there are on average about one MHW or MCS per year in the Tasman Sea.

L241-245: I appreciate this approach, which is the correct one since waves are by definition variability relative to the mean state. This is an ongoing discussion in the field (e.g., Jacox et al., Nature, 2019), and this study is a good example of one reason to remove trends (so that mechanisms can be separated from long-term warming). I disagree with the statement at L243-245 though; after detrending there can still be changes in intensity/frequency, which would come from changes in the SST anomaly distribution (e.g., increasing/decreasing variance).

I agree that there could be changes in intensity/frequency in MHW computed from detrended data – hence the word 'perhaps'. On reflection, I think that this statement is confusing and have deleted it.

Please see new second paragraph in the Discussion about trends

L247-253: I think this paragraph should be cut – it doesn't really fit and doesn't draw on the results at all.

I have deleted this paragraph.

L273-283: This explanation isn't clear to me, particularly because the anomalous winds are not apparent. From the results presented here, the main driver of MHW/MCS appears to be the surface heat flux anomalies (which the author says are dominated by the latent heat flux). Please clarify the explanation.

Previously I was computing the meridional and zonal components separately, now I compute wind stress as a scalar and compute anomalies appropriately. This has changed the Hovmuller diagrams a little and there is new text to go with them.

Discussion: Somewhere it would be good to discuss the implications of the findings. e.g., might there be predictability associated with the atmospheric pattern.

The last paragraph is new and discusses the implications of this work " Finally, the canonical MHW/MHS derived in here are, by virtue of their definition, broad-scale events centred on the Tasman Sea. Individual MHW/MCS show considerable variability in both spatial extent and intensity (**Error! Reference source not found.**) so that while the large-scale atmospheric

patterns appear to set up the MHW/MCS, the local response is driven by smaller-scale variability. Nevertheless, the results of this work suggest that the key to making accurate predictions of MHW/MCS globally lies in understanding the causes of, and being able to predict, the W4 atmospheric wave stalling and intensification.”

L307-308: Were the 90th percentile anomalies the same for the whole year or were they seasonally dependent (as in Hobday et al)?

The 90th percentiles were the same for the whole year. Seasonally-dependent 90th percentiles tend to have minima in winter so that potentially a 0.1°C (as an example) excursion might be considered a Category 3 heatwave in winter, but normal in summer. My opinion is that it is better to have a time-independent threshold value.

L313-316: Inconsistency here: is SSTAS the area averaged SST (as stated in 313-315) or the area averaged normalized SST anomaly (as suggested in 315-316)?

This has been corrected to state that it is the area-averaged SST anomaly.

L316-317: What is the justification for a 90 day separation? I haven't seen anything like that before and I can't think of a reason for it. And if there are events within 90 days of each other how do you choose which to keep?

The 90-day separation was to avoid the occasional instance where there would otherwise have been two back-to-back MHW (or MCS). Given the large-scale nature of the events, these double events would be forced by the same atmospheric conditions, and I did not want such double events to bias the statistics. I.e., I wanted the ensemble of MHW (in this case 22 events) to all be independent events. The 90 days (one season) was arbitrarily chosen to be more than a month (which is about the Eulerian time scale for the ocean) and less than a year.

The first event was chosen.

L326-327: Please provide a brief description of the methodology that does not require reading Santer et al.

I now include a brief description in the Methods Section. “The significance of the slope of linear regressions was calculated following Santer et al. (2000), where the trend is considered significantly different from zero if the ratio of the estimated trend and its standard error, t_b , falls above the Student's t test value given a confidence level and number of degrees of freedom. The confidence level was taken to be 95% and the number of degrees of freedom was computed as the record length minus 2, assuming every 15th daily satellite estimate was independent.”

- Bâki Iz, H. (2018). Is the global sea surface temperature rise accelerating? *Geodesy and Geodynamics*, 9(6), 432-438. doi:10.1016/j.geog.2018.04.002
- Chiswell, S. M., & O'Callaghan, J. M. (2021). Long-term trends in the frequency and magnitude of upwelling along the West Coast of the South Island, New Zealand, and the impact on primary production. *New Zealand Journal of Marine and Freshwater Research*, 1-22. doi:10.1080/00288330.2020.1865416
- Feng, M., Caputi, N., Chandrapavan, A., Chen, M., Hart, A., & Kangas, M. (2021). Multi-year marine cold-spells off the west coast of Australia and effects on fisheries. *Journal of Marine Systems*, 214. doi:10.1016/j.jmarsys.2020.103473
- Hobday, A., Oliver, E., Sen Gupta, A., Benthuysen, J., Burrows, M., Donat, M., et al. (2018). Categorizing and Naming Marine Heatwaves. *Oceanography*, 31(2). doi:10.5670/oceanog.2018.205
- Jacox, M. G. (2109). Marine heatwaves in a changing climate. *Nature*, 571(7766), 485-487. doi:10.1038/d41586-019-02196-1
- Oliver, E. C. J., Donat, M. G., Burrows, M. T., Moore, P. J., Smale, D. A., Alexander, L. V., et al. (2018). Longer and more frequent marine heatwaves over the past century. *Nature Communications*, 9(1), 1324. doi:10.1038/s41467-018-03732-9
- Santer, B. D., Wigley, T. M. L., Boyle, J. S., Gaffen, D. J., Hnilo, J. J., Nychka, D., et al. (2000). Statistical significance of trends and trend differences in layer-average atmospheric temperature time series. *Journal of Geophysical Research: Atmospheres*, 105(D6), 7337-7356. doi:10.1029/1999jd901105

REVIEWERS' COMMENTS

Reviewer #1 (Remarks to the Author):

Happy with the authors responses to my comments

Reviewer #2 (Remarks to the Author):

The text and figures are much improved. Below are my comments and suggestions on the revised version.

L27: The revision here missed my original point. Saying "not only... but also..." gives the impression that the warming trend and increased MHWs are two effects that are compounding. In reality they are the same thing – when the ocean warms and you don't account for the trend, MHW thresholds are exceeded more often. It would be more accurate to say something along the lines of: SST is warming (refs 1,2), so increases in exceedance of historical MHW thresholds (refs 3,4) reflect the warming trends, not changes in the characteristics of MHWs themselves (ref 5).

L47-48: Reduced ocean mixing associated with a persistent ridge of higher than normal atmospheric pressure

L66: MHWs being most common in a particular season is dependent on definitions. It may be true with a fixed MHW threshold, but many studies use a seasonally-varying MHW threshold, which results in MHWs being distributed throughout the year.

L78: I would not call reanalyses "satellite-derived". They are models that assimilate observations, including those from satellites.

L157-159: I would cut this out, as I don't think there's any physical basis for constructing such an index.

L182-183: On average, Tasman Sea MHW/MCS occur with warm/cold anomalies in these other regions, but not MHW/MCS. SSTA in the other regions would be >1 if on average they were MHW/MCS.

L185: For this approach, suggest using "composite" instead of "canonical"

L206: typo

L233: should be positive wind stress anomaly I think

L238: I would rephrase as I don't think this first sentence captures the main findings (which I would say is the atmospheric anomaly patterns surrounding Tasman Sea MHW/MCS).

L251: But also agree with other researchers who have removed the trend (examples below). These don't all have to be cited, but the point is there is precedent for removing the trend (and reasons why outlined in the earlier cited Jacox 2019).

Holbrook, N. J. et al. A global assessment of marine heatwaves and their drivers. *Nat. Commun.* 10, 2624 (2019)

Jacox, M. G. et al. Thermal displacement by marine heatwaves. *Nature*, 584, 82–86

Scannell, H. A., Pershing, A. J., Alexander, M. A., Thomas, A. C. & Mills, K. E. Frequency of marine

heatwaves in the North Atlantic and North Pacific since 1950. *Geophys. Res. Lett.* 43, 2069–2076 (2016).

L258: It would be very easy to test this out. If you want to include this text there's no need to speculate, instead try the different trend removals and see how things change.

L296: Have you looked at the SSH field?

L317-321: Again, I would remove the modified index

L327-330: Perhaps regionally this is true, but not globally

L343: I'm still not sold on the justification for the 90-day separation, but in any case it should be expanded upon here so the reader can judge for themselves.

L358: Is 15d consistent with autocorrelation scales in the region?

L361: typo

Fig. 5: Would be good to have the colors not saturate (i.e., expand the range of plotted values for SSTA)

Fig. 6: labels for axes and colorbars are overlapping

Mike Jacox
NOAA

REVIEWERS' COMMENTS

Reviewer #1 (Remarks to the Author):

Happy with the authors responses to my comments

Reviewer #2 (Remarks to the Author):

The text and figures are much improved. Below are my comments and suggestions on the revised version.

L27: The revision here missed my original point. Saying “not only... but also...” gives the impression that the warming trend and increased MHWs are two effects that are compounding. In reality they are the same thing – when the ocean warms and you don’t account for the trend, MHW thresholds are exceeded more often. It would be more accurate to say something along the lines of: SST is warming (refs 1,2), so increases in exceedance of historical MHW thresholds (refs 3,4) reflect the warming trends, not changes in the characteristics of MHWs themselves (ref 5).

Since the aim of this first paragraph is only to introduce MHW, I have simplified it by removing the direct reference to ocean warming, and defer discussion of the relative roles of MHW and long-term warming until later. I now say -

“It has been suggested that marine heat waves (MHW) have become stronger and or more frequent over the last century^{1,2} (although it has been pointed out that such suggestions may not take into account long-term warming³). MHW are sometimes considered as good analogues for possible future oceans⁴, ...”

L47-48: Reduced ocean mixing associated with a persistent ridge of higher than normal atmospheric pressure

I have incorporated these words directly, and now say “Bond, et al. ¹¹ suggested MHW in the north-east Pacific Ocean were primarily driven by reduced ocean mixing associated with a persistent ridge of higher than normal atmospheric pressure.”

L66: MHWs being most common in a particular season is dependent on definitions. It may be true with a fixed MHW threshold, but many studies use a seasonally-varying MHW threshold, which results in MHWs being distributed throughout the year.

I agree, and since MHW have not been defined at this stage of the paper, I have removed the words “(i.e. when MHW are most common)”,

L78: I would not call reanalyses “satellite-derived”. They are models that assimilate observations, including those from satellites.

I have removed the words “satellite-derived”, and now say “...use global reanalyses of...’

L157-159: I would cut this out, as I don't think there's any physical basis for constructing such an index.

This has been removed.

L182-183: On average, Tasman Sea MHW/MCS occur with warm/cold anomalies in these other regions, but not MHW/MCS. SSTA in the other regions would be >1 if on average they were MHW/MCS.

I accept this point, and now say "MHW/MCS in the Tasman Sea co-occur with corresponding warm and cold anomalies in the Indian ..."

L185: For this approach, suggest using "composite" instead of "canonical"

I prefer to use the term "canonical" because it suggests a standard event. There is precedent for this, since canonical has long been used to describe the standard El Niño
For example – see –

ENSO regimes: Reinterpreting the canonical and Modoki El Niño K. Takahashi, Montecinos, K. Goubanova, and B. Dewitte, GEOPHYSICAL RESEARCH LETTERS, VOL. 38, L10704, doi:10.1029/2011GL047364, 2011

Yeh S. W., Kug J. S., Dewitte Boris, Kwon M. H., Kirtman B. P., Jin F. F. (2009). ENiño in a changing climate. Nature, 461 (7263), p. 511-514. ISSN 0028-0836.

L206: typo

Fixed.

L233: should be positive wind stress anomaly I think

Indeed ! Thanks...

L238: I would rephrase as I don't think this first sentence captures the main findings (which I would say is the atmospheric anomaly patterns surrounding Tasman Sea MHW/MCS).

I have rewritten this paragraph. I now say –

"In summary, when SST anomalies are defined with respect to a linear trend to take into account long-term change, and normalised by their 90th percentile to take into account local variability, about 20 MHW and MCS each occurred in the Tasman Sea between 1982 and 2020, but these are not uniformly distributed in time. There is little evidence that MHW are increasing in either frequency or intensity in the Tasman Sea, although their duration may be increasing. MCS showed a significant increase in strength, but appear to be decreasing in frequency. The main findings are that MHW/MCS in the Tasman Sea are part of a global phenomenon driven by a wavenumber-4 (W4) atmospheric forcing, and there is no clear relationship between MHW/MCS and SAM or SOI.

L251: But also agree with other researchers who have removed the trend (examples below). These don't all have to be cited, but the point is there is precedent for removing the trend (and reasons why outlined in the earlier cited Jacox 2019).

I now say "In this respect, we differ from researchers who do not remove a trend and conclude that MHW are increasing in intensity{Oliver, 2018 #1872;Frölicher, 2018 #2084}, but agree with other researchers who also remove the trend{Jacox, 2020 #2081}"

L258: It would be very easy to test this out. If you want to include this text there's no need to speculate, instead try the different trend removals and see how things change.

I now say "Had a trend that allowed for accelerated heating been removed, these two conclusions would have been different. If the 10-year running mean had been removed instead of a linear trend, the increase in MHW duration would have been shorter and less significant (4 d decade⁻¹, p=0.04, vs 7 d decade⁻¹, p=0.01). This points not only to the importance...

L296: Have you looked at the SSH field?

No – there are no westward-propagating anomalies in SST or heat flux, so even if Rossby waves appeared in SSH, they would apparently have little impact on the SST, heat flux. It would be interesting to do such an analysis, but I think it is beyond the scope of this paper.

L317-321: Again, I would remove the modified index

This has been removed

L327-330: Perhaps regionally this is true, but not globally

I now say " ... the local response in the Tasman Sea is driven by ..."

L343: I'm still not sold on the justification for the 90-day separation, but in any case it should be expanded upon here so the reader can judge for themselves.

I now say "This separation requirement was chosen so that the occasional case where SST_{TAS} fluctuated around the limits over several weeks counted as one event to avoid biasing the mean statistics towards any particular year.

L358: Is 15d consistent with autocorrelation scales in the region?

In principle the time between independent estimates should be taken from the first zero in the autocorrelation. However, the autocorrelation function never goes to zero. In this case, I estimated the time between independent estimates to be the lag when the autocorrelation-squared (r^2) was 50%, i.e., half the variance is independent.

I now say “assuming every 15'th daily satellite estimate was independent (15 d was chosen as this is the lag when the SST_{TAS} autocorrelation-squared is 0.5).”

L361: typo
fixed

Fig. 5: Would be good to have the colors not saturate (i.e., expand the range of plotted values for SSTA)

I have expanded the range as suggested to minimise the saturation.

Fig. 6: labels for axes and colorbars are overlapping
This has been fixed.

Mike Jacox
NOAA